# Novel Y RNA-Derived Fragments Can Differentiate Canine Hepatocellular Carcinoma from Hepatocellular Adenoma

**DOI:** 10.3390/ani13193054

**Published:** 2023-09-28

**Authors:** Norio Ushio, Md Nazmul Hasan, Mohammad Arif, Naoki Miura

**Affiliations:** 1United Graduate School of Veterinary Science, Yamaguchi University, 1677-1 Yoshida, Yamaguchi 753-0841, Japan; ushio4762@vanilla.ocn.ne.jp; 2Joint Graduate School of Veterinary Medicine, Kagoshima University, 1-21-24 Korimoto, Kagoshima 890-0065, Japan; nazmulriaj.bau.vet@gmail.com (M.N.H.); mdarif38515@bau.edu.bd (M.A.); 3Clinical Veterinary Division, Faculty of Veterinary Medicine, Airlangga University, Mulyorejo, Surabaya 60115, Indonesia

**Keywords:** Y RNA-derived fragments, canine hepatocellular carcinoma, canine hepatocellular adenoma

## Abstract

**Simple Summary:**

Hepatocellular carcinoma (HCC) is difficult to distinguish from hepatocellular adenoma (HCA) in dogs, and HCC may develop from HCA, according to recent reports. Therefore, urgent research is needed to establish a reliable biomarker for the early detection of these hepatic diseases. Noncoding RNAs (ncRNAs) could be a very useful tool for diagnosing hepatic diseases. Y RNA is a regulatory RNA type with a sequence of 80–110 nucleotides. In this study, we investigated novel Y RNA-derived fragments, namely Y RNA, which we previously investigated in canine mammary gland tumors and found that they could differentiate between benign and malignant tumors. Accordingly, we decided to investigate Y RNA in canine HCC and HCA, which has not been attempted before in either humans or dogs. We report that Y RNA can discriminate canine HCC from HCA and could be a promising biomarker for diagnosing canine HCC and HCA.

**Abstract:**

Hepatocellular carcinomas (HCC) are common tumors, whereas hepatocellular adenomas (HCA) are rare, benign tumors in dogs. The aberrant expression of noncoding RNAs (ncRNAs) plays a pivotal role in HCC tumorigenesis and progression. Among ncRNAs, micro RNAs have been widely researched in human HCC, but much less widely in canine HCC. However, Y RNA-derived fragments have yet to be investigated in canine HCC and HCA. This study targeted canine HCC and HCA patients. We used qRT-PCR to determine Y RNA expression in clinical tissues, plasma, and plasma extracellular vesicles, and two HCC cell lines (95-1044 and AZACH). Y RNA was significantly decreased in tissue, plasma, and plasma extracellular vesicles for canine HCC versus canine HCA and healthy controls. Y RNA was decreased in 95-1044 and AZACH cells versus normal liver tissue and in AZACH versus 95-1044 cells. In plasma samples, Y RNA levels were decreased in HCC versus HCA and Healthy controls and increased in HCA versus Healthy controls. Receiver operating characteristic analysis showed that Y RNA could be a promising biomarker for distinguishing HCC from HCA and healthy controls. Overall, the dysregulated expression of Y RNA can distinguish canine HCC from HCA. However, further research is necessary to elucidate the underlying Y RNA-related molecular mechanisms in hepatocellular neoplastic diseases. To the best of our knowledge, this is the first report on the relative expression of Y RNA in canine HCC and HCA.

## 1. Introduction

Hepatocellular adenoma (HCA) and hepatocellular carcinoma (HCC) can occur in both canine and human patients [1]. HCAs are rare, benign tumors that derive from proliferating hepatocytes, whereas HCCs are common, malignant tumors that can develop from HCAs [2]. HCC accounts for 50–59.4% of hepatic tumors in dogs [3] and is the sixth most common cancer in humans worldwide [4]. Canine HCC frequently occurs in patients from the age of 10 years and these tumors are mainly common in males [5]. Distinguishing HCA from HCC can be complicated [6,7], but correct tumor identification is crucial because the indicated treatment and prognosis differ between these two tumor types. Therefore, there is a need for a minimally invasive diagnostic technique for differentiating HCA from HCC on a molecular basis for canine patients.

Noncoding RNAs (ncRNAs) are potentially implicated in hepatocellular tumorigenesis and may serve as a diagnostic marker for these tumors [8,9]. ncRNA categories encompass diverse transcripts, including miRNAs, long noncoding RNAs, and other RNA-like Y RNA fragments. Studies on miRNAs in human HCA are limited [2,10], and there is only one report on miRNAs involved in canine HCA [11]. miRNA involvement in HCC growth has been the subject of in-depth research in dogs and humans [11,12,13,14,15]. For example, our group has previously reported miR-1 dysregulation in canine HCC [12]. However, none of the reports on ncRNAs in canine liver tumors have addressed Y RNA.

Y RNA is first reported in patients with systemic lupus erythematosus in 1981 [16]. Despite being highly conserved molecules, Y RNAs exist in all vertebrate species [17], and the number of Y RNA transcripts varies between species [18]. Y RNAs are a type of regulatory RNA that have a sequence of 80–110 nucleotides [19]. They are identified by a stem-loop structure formed by complementary 5′ and 3′ ends [18]. Y RNAs may follow the miRNA’s biogenesis pathways due to having the stem-loop structure of both Y RNAs and miRNAs [18]. Another study suggests that Y RNAs do not enter the miRNA biogenesis pathway and also do not bind to argonaut complex protein [20]. Y RNAs are transcribed by the enzyme RNA polymerase III. These RNAs are bound to the polyuridine tail of the La protein, also known as small RNA binding exonucleolytic protection factor. This binding ensures nuclear retention and safeguards them from degradation [21]. Additionally, Y RNAs are also bound to RO60, also known as SSA, which promotes nuclear export and makes them more stable [22]. Y RNA-derived fragments (YRFs) are formed as a result of the partial breakdown of Y RNAs during apoptosis, which is carried out via the caspase-3-dependent pathway [20]. YRFs have been detected in both normal and cancerous tissues [23]. 

The dysregulation of Y RNAs may contribute to the development of tumors, affect cell growth, and promote inflammation [24]. Y RNAs are crucial in initiating DNA replication, maintaining RNA stability, and responding to cell stress [18,25]. Y RNAs are responsible for cellular processes such as cell proliferation [18]. Y RNAs and YRFs might be involved in signaling or a gene regulation function [26,27]. Y RNAs have not previously been investigated in human or canine HCC and HCA.

Y RNAs have been found in substantial amounts in plasma and serum from human patients [28,29], other biofluids [30], and extracellular vesicles [31]. Y RNAs have been established as reliable diagnostic biomarkers for a range of human cancers, including prostate [32] and bladder [33] cancers, melanoma [28], head, and neck squamous cell carcinoma (HNSCC) [24], breast cancer [29], lung cancer [34] and clear cell renal cell carcinoma [35]. Regarding evidence from dogs, our group has found decreased Y RNA-fragment expression in canine mammary gland tumors [36].

Extracellular vehicles (EVs) are of potential interest for the quantification of Y RNA fragments and other ncRNAs. They are small structures released by cells to facilitate the transportation of vital components, such as DNA, RNA, and proteins, for effective intercellular communication [37]. EV-derived ncRNAs have great significance for the early diagnosis of HCC due to their presence in circulation at an early stage of the disease, and they also have implications for any drug delivery system used in the treatment of HCC [38,39]. Recent studies have shown that EV-derived Y RNA is abundant in human small-cell lung cancer [40], melanoma [41], and brain tumors [42]. However, EV-derived Y RNA has yet to be studied in either canine or human HCC or HCA. 

Similar gene expression patterns, such as the significance of *TGF-beta*, seem to be evident in the development of HCC in dogs and humans [43]. That is why exploring the role of Y RNA presents a promising avenue for gaining significant insights into the development of hepatic diseases. 

Accordingly, in this study, we aimed to determine relative Y RNA expression levels in dogs with HCC and HCA using qRT-PCR analysis, targeting tumor tissues, plasma, and plasma EVs from clinical samples, and HCC cell lines, to evaluate Y RNAs as diagnostic biomarkers for these two types of liver tumor in dogs. 

## 2. Materials and Methods

### 2.1. Study Population (Clinical Samples)

The clinical samples evaluated in this study had been obtained from a population of 28 dogs (age range: 8–14 years) diagnosed histo-pathologically with HCA (*n* = 15) or HCC (*n* = 13) by a veterinary pathologist when undergoing surgery at the Kagoshima University Veterinary Teaching Hospital or an affiliated clinic, between September 2012 and December 2022. The owner of each dog gave informed consent the use of samples in this research. Samples were also collected from nine healthy adult laboratory beagle dogs to include as healthy controls in the evaluation provided by Shin Nippon Biomedical Laboratories, Ltd. (Tokyo, Japan) [12].

Tumor tissue samples were collected at the time of surgery from the clinical patients, and biopsy samples were collected from the livers of healthy controls. Plasma samples were obtained from a subset of the study population (*n* = 20; Healthy controls: *n* = 6; HCA: *n* = 5; HCC *n* = 9). Full details of the HCA and HCC patients are summarized in Table 1. Tissue samples were immersed in RNA immediately after collection and stored at −80 °C for long-term preservation. Blood samples were collected in anticoagulant-treated tubes (Terumo Venoject tubes 3.2% sodium citrate) and centrifuged at 3000× *g* for 10 min to remove the cell debris. The plasma samples were separated and centrifuged again at 16,000× *g* at 4 °C to remove the debris. The supernatant was transferred to new Eppendorf tubes and stored at −80 °C as plasma samples.

### 2.2. Cell Lines and Cell Culture

In this study, we evaluated two HCC cell lines, 95-1044 (a fast-proliferating cell line) and AZACH (an intermediate-proliferating cell line) [12,44]. Cell lines were preserved using a CultureSure freezing medium and stored in liquid nitrogen (Wako Pure Chemical Industries, Ltd., Osaka, Japan). D-MEM medium (Sigma-Aldrich, St. Louis, MO, USA), 5% fetal bovine serum (Thermo Fisher Scientific, Waltham, MA, USA), 5% L-glutamine (Sigma-Aldrich), and 3.5 μg/mL spectinomycin (Sigma-Aldrich) were used to culture the cells. All cells were cultured in a humidified incubator with 5% CO_2_ at 37 °C. Cold phosphate-buffered saline (PBS) and 0.25% trypsin or 0.1% EDTA were applied for detachment of the cells. Cells were counted using an automated cell counter (LUNAII, Logos, Anyang, South Korea). 

### 2.3. EV Isolation

The Total Exosome RNA and Protein Isolation Kit (Invitrogen, Waltham, MA, USA, Thermo Fisher Scientific) was used to isolate EVs from plasma, following the manufacturer’s protocol. In brief, 300 μL plasma samples were mixed with a half volume of 1X PBS, 90 μL of exosome precipitation reagent was then added, and the resultant mixture was vortexed thoroughly and centrifuged at 10,000× *g* for 5 min. The supernatant was discarded, and the tube was centrifuged again at 1000× *g* for 30 s to remove the residual reagent. Finally, the pellet was reconstituted in 150 μL 1X PBS and stored at −80 °C for further analysis.

### 2.4. RNA Isolation of Clinical Samples and HCC Cell Lines

A mirVana^TM^ RNA Isolation Kit (Thermo Fisher Scientific, Waltham, MA, USA) was used to extract total RNA from tissues and cells in accordance with the manufacturer’s protocol. A mirVana PARIS Kit (Thermo Fisher Scientific) was used to isolate total RNAs from plasma samples and EVs. Before extraction, 5 μL (5 femtomoles) of synthetic cel-miR-39 was added to every plasma and plasma EV sample for normalization. Briefly, each tissue sample or the relevant HCC cell preparation was mixed with the required amount of lysis buffer. A 300 µL aliquot of each plasma sample was mixed with an equivalent amount of 2× denaturation solution. A 1:10 ratio of a miRNA homogenate additive was added to the tissue or cell lysate, then kept on ice for 10 min. An amount of 600 µL Acid-phenol:chloroform (Ambion^®^) was added to the tissue, cell lysate, or plasma, with subsequent thorough vortex-mixing and then centrifugation at 15,000× *g* for 5 min at room temperature. The supernatant was then collected carefully in an Eppendorf tube, to which a 1.25-fold amount of molecular-grade ethanol (99.9% in purity) was added (and the amount recorded), and the tube contents were filtered using centrifugation. In the final step, total RNA was obtained as sediment in the tube using an elution solution pre-heated to 95 °C. The NanoDrop 2000c spectrophotometer was used to measure the concentration of total RNA (Thermo Fisher Scientific). To evaluate the quality and integrity of RNA, an Agilent 2100 Bioanalyzer was utilized (Agilent Technologies, Santa Clara, CA, USA). The cells and tissues had RNA Integrity Numbers ranging from 8.5 to 9.5.

### 2.5. ncRNAs Selection and qRT-PCR

Y RNA was selected based on a previously published NGS dataset (SRA: PRJNA716131) for canine mammary gland tumors [36]. The qRT-PCR protocol was described previously [45,46,47]. Briefly, 2 ng (for tissues and cell lines) or 1.25 µL (for plasma and plasma EVs) of total RNA were reverse transcribed to cDNA using the TaqMan MicroRNA Reverse Transcription Kit (Thermo Fisher Scientific) with a T100 thermal cycler, following the manufacturer’s protocol. For qRT-PCR, a TaqMan First Advanced Master Mix Kit and a Quant Studio 3 real-time PCR system (Thermo Fisher Scientific) were applied. Each experiment was conducted twice to ensure accuracy. To evaluate the expression level, the 2^−ΔΔCT^ method was used. RNU6B was used as an internal control for tissues and plasma, miR-16 was for the plasma, and miR-186 was for EVs [48]. The TaqMan primer sequences are as follows; 5′-GGCTGGTCCGAGTGCAGTGGTGCTTAC-3′ YRNA fragments (Ensembl ID: ENSCAFT00000034244.1).

### 2.6. Statistical Analysis

GraphPad Prism 9 (https://www.graphpad.com/, accessed on 1 August 2023) was used for statistical analysis and graph visualization. A Mann–Whitney U test and a one-way analysis of variance (ANOVA) followed by the Kruskal-Wallis test were used to assess the qRT-PCR results where applicable. ROC curves and AUCs were plotted using Wilson/Brown method. A *p*-value < 0.05 was considered statistically significant.

## 3. Results

### 3.1. Expression of Y RNA Using qRT-PCR

#### 3.1.1. Relative Expression in the Clinical Tissue Samples

The relative expression of Y RNA was investigated in HCA and HCC tissue samples. YRNA was significantly decreased in HCC (fold change (FC) = 0.43, *p* = 0.008) versus healthy controls (Figure 1). In addition, Y RNA was preferentially decreased in HCC (FC = 0.39, *p* = 0.001) versus HCA. However, the Y RNA expression level did not significantly differ between healthy controls and HCA. Thus, the expression profile for Y RNA in HCC differed to those in healthy controls and HCA.

#### 3.1.2. Relative Expression in Plasma

We evaluated the expression of the selected Y RNA in plasma samples. Y RNA was significantly decreased in HCC (FC = 0.02, *p* = 0.002) and significantly increased in HCA (FC = 2.50, *p* = 0.002) versus Healthy controls (Figure 2). Furthermore, Y RNA expression was decreased in HCC (FC = 0.009, *p* = 0.030) versus HCA. Taken together, our findings indicate that Y RNA expression may differentiate HCC and HCA from Healthy controls and HCC from HCA.

#### 3.1.3. Relative Expression in Plasma EVs

In plasma, the relative Y RNA expression was significantly decreased in HCC (FC = 0.21, *p* = 0.001) versus Healthy controls (FC = 0.21, *p* = 0.001), and HCA (FC = 0.06, *p* = 0.001) (Figure 3). However, Y RNA expression did not significantly differ between HCA and Healthy controls. Thus, Y RNA expression could distinguish HCC and HCA from Healthy controls. The Y RNA expression profile in plasma was consistent with that in clinical tumor tissue samples. 

#### 3.1.4. Relative Expression in Canine HCC Cell Lines

The relative expression of Y RNA was evaluated in fast-proliferative 95-1044 and intermediate-proliferating AZACH cell lines. Y RNA was significantly decreased in 95-1044 (FC = 0.03, *p* = 0.0002) and AZACH (FC = 0.24, *p* = 0.0007) cells versus normal liver tissue (Figure 4). In addition, Y RNA was significantly decreased in 95-1044 cells (FC = 0.15, *p* = 0.004) versus AZACH cells. Our results thus suggest that Y RNA expression is substantially decreased in fast-proliferative HCC cell lines, which is consistent with the results for clinical tumor tissue samples.

### 3.2. Diagnostic Value of Y RNA 

Receiver operating characteristics (ROC) curves and areas under the curve (AUCs) were generated to investigate the diagnostic value of Y RNA. In plasma analyses, Y RNA yielded AUCs of 0.920 (*p* = 0.028) and 1.00 (*p* = 0.004) for HCA and HCC, respectively, when evaluated against Healthy controls (Figure 5A,B). Y RNA also differentiated HCC (AUC = 1.00, *p* = 0.004) from HCA in plasma samples (Figure 5C). In plasma EV analyses, Y RNA significantly distinguished HCC (AUC = 0.963, *p* = 0.003) from Healthy controls (Figure 5D) and from HCA (AUC = 1.00, *p* = 0.005; Figure 5E); however, it could not distinguish HCA from Healthy controls (AUC = 0.833, *p* = 0.088; Appendix A). In summary, Y RNA could discriminate HCC and HCA from Healthy controls and HCC from HCA in plasma and plasma EVs. 

## 4. Discussion

ncRNAs play a pivotal role in HCC development and progression, and evidence exists to support their utility as diagnostic and prognostic biomarkers for this disease [49,50,51]. Among ncRNAs, miRNAs have showed similar expression patterns in extensive studies on human and canine HCC [11,12,13,14,15]. miRNAs are less commonly studied in human HCA [2,10], and have featured in only one study in canine HCA, in which dysregulation was found [15]. In contrast, Y RNA-derived fragments have not previously been studied in human or canine HCC or HCA, and here we report original findings (to our knowledge) on Y RNA expression in these two types of liver tumors in canine patients. 

In key findings, Y RNA expression was significantly decreased in canine HCC tumor tissue versus healthy controls, and HCA tumor tissue, and the same pattern was noted in plasma EV samples. In plasma samples, Y RNA was significantly decreased in canine HCC and significantly increased in HCA versus Healthy controls. We also investigated Y RNA in two canine HCC cell lines and found that it was significantly decreased in fast-proliferating 95-1044 cells and intermediate proliferating AZACH cells versus normal liver tissue. 

The expression pattern of Y RNA in HCC cell lines was similar to that in clinical tissues. Y RNA was found to be decreased in HCC and in 95-1044 and AZACH cells, versus the control liver samples. This finding is interesting because results for the HCC cell lines reflected those in clinical tissue HCC samples, in comparisons against the same control liver samples. ROC analyses revealed that Y RNA could distinguish HCC from the Healthy controls and HCA patients in plasma and plasma EV analyses. 

Altered Y RNA and YRFs expression levels are potentially implicated in carcinogenesis, and there is evidence that they act as diagnostic and prognostic biomarkers for several cancers [24,32,52]. Oncologists focusing on the human prostate have found that RNY1, RNY3, RNY4, and RNY5 are downregulated in prostate adenocarcinoma versus normal tissue and benign prostate hyperplasia [32]. These Y RNAs (RNY1, RNY3, RNY4, and RNY5) are reportedly similarly downregulated in human bladder cancer versus normal urothelial bladder tissue and act as a prognostic indicator for this condition [33]. RNY3P1, RNY4P1, and RNY4P25 show significantly higher expression in stage 0 human melanoma than at more advanced stages [28]. YRNA1 and YRNA5 are downregulated in human HNSCC, for which YRNA1 is regarded as a potential biomarker [24]. Deep sequencing and bioinformatics analysis-based study has reported that dysregulated Y RNAs are also abundant in serum of human breast cancer patients [29]. YRNA-RNY1 is downregulated in human lung cancer patients compared to normal patients, whereas YRNA-RNY1 is found to be upregulated in lung cancer patients suffering from tuberculosis compared to normal controls [34]. In clear cell renal cell carcinoma, hY3 and hY4 show altered expression compared to normal renal tissue [35]. hY1 and hY3 RNA are more highly abundant and upregulated in colon cancer patients than in healthy controls [53]. A set of Y RNAs (hY1, hY3, and hY4) are shown as increased in human cervix cancer [53]. We are currently compiling evidence that Y RNA is substantially decreased in metastasized canine mammary gland tumors versus those classified as benign mixed tumors [36]. A recent study revealed that hY4 RNA fragments are upregulated in human small-cell lung cancer-derived EVs and inhibit tumor development by inhibiting MAPK/NF-_k_B signaling [40]. Deep sequencing-based studies have shown that EV-derived Y RNAs are abundant in human melanoma [41]. Y RNAs are also found to be abundant in human brain tumor-derived EVs [42]. RNY4 fragments are highly abundant in non-Hodgkin lymphoma-derived EVs [54]. hY5 RNA is shown to be enriched in blood cancer-derived EVs (K562 cells and myelogenous leukemia) [55]. Overall, the findings in this study are consistent with several reports on human cancer and canine MGT, indicating that Y RNA expression is decreased in malignant tumors (such as canine HCC) relative to benign tumors (such as canine HCA) and healthy controls.

A recent study revealed that canine HCA transforms into HCC, which means recurrence may occur [56]. Therefore, this study demonstrated that Y RNA has a high potential for distinguishing canine HCA from HCC. We believe these findings provide insights into comprehending the knowledge of differential diagnoses among hepatic diseases. 

The functional roles of Y RNA in canine HCC and HCA and its participation in the relevant underlying molecular mechanisms have yet to be fully elucidated. Here, we have demonstrated the aberrant expression of this ncRNA in canine HCC and HCA patients. We posit that Y RNA might be involved in cancer malignancy through its downregulated expression in HCC. Y RNA could be a biomarker distinguishing malignant tumors (HCC) from benign tumors (HCA) and tumor-free patients. However, this study still has some limitations. First, our study sample was relatively small. We need to validate Y RNA in a large cohort sample to strengthen our findings further. Second, the roles of Y RNA in canine HCC development need to be investigated in a bio-functional study.

Pet dogs have great potential utility for comparative oncology clinical trials, partly because they maintain an intact immune system and experience natural co-evolution of the tumor microenvironment [57]. Humans and dogs are known to develop cancer through aberrations occurring for the same genes [58]. Therefore, this study has great potential to enhance our understanding of the expression of Y RNA in hepatic diseases.

## 5. Conclusions

To our knowledge, this is the first report on Y RNA in canine HCC and HCA. This ncRNA has distinctive characteristics and differentiates malignant tumors (HCC) from benign tumors (HCA). Notably, its expression pattern is consistent across clinical samples and cell lines. We thus consider that Y RNA has promising potential for differentiating HCC from HCA. Our findings provide significant insights into how Y RNA contributes to the progression of hepatic disease in dogs. Further research is required to fully elucidate the role of Y RNA in the development and progression of canine HCC and HCA.

## Figures and Tables

**Figure 1 animals-13-03054-f001:**
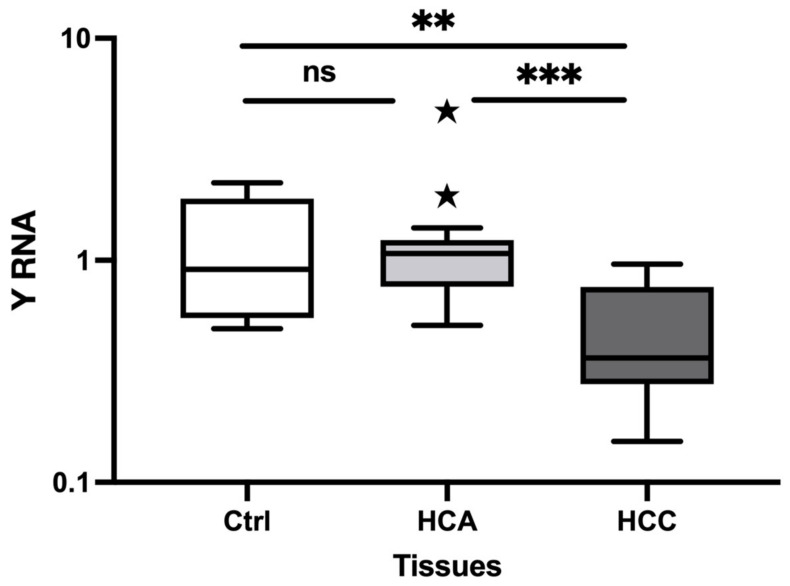
Relative expression of Y RNA in HCA and HCC tissue samples using qRT-PCR. The relative expression level of Y RNA in HCA (*n* = 15) and HCC (*n* = 13) versus normal liver tissue (*n* = 9). The *Y*-axis represents relative noncoding RNA expression levels in log10 units. One-Way ANOVA (nonparametric) was performed, followed by the Kruskal–Wallis and Mann–Whitney tests. Differences were considered significant when the *p*-value was < 0.05 (** *p* < 0.01, *** *p* < 0.001). ns: not significant; Ctrl: Control; HCA: Hepatocellular adenoma; HCC: Hepatocellular carcinoma.

**Figure 2 animals-13-03054-f002:**
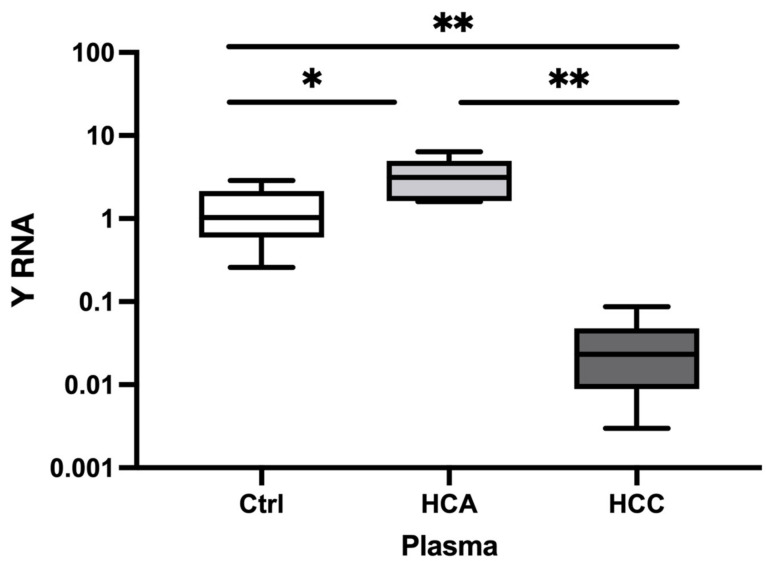
Relative expression of Y RNA in HCA and HCC plasma. The relative expression level of Y RNA in plasma HCA (*n* = 5) and HCC (*n* = 7) versus Healthy controls (*n* = 6). The *Y*-axis represents relative noncoding RNA expression levels in log10 units. One-Way ANOVA (nonparametric) was performed, followed by the Kruskal–Wallis and Mann–Whitney tests. Differences were considered significant when the *p*-value was < 0.05 (* *p* < 0.05, ** *p* < 0.01) Ctrl: control; HCA: Hepatocellular adenoma; HCC: Hepatocellular carcinoma.

**Figure 3 animals-13-03054-f003:**
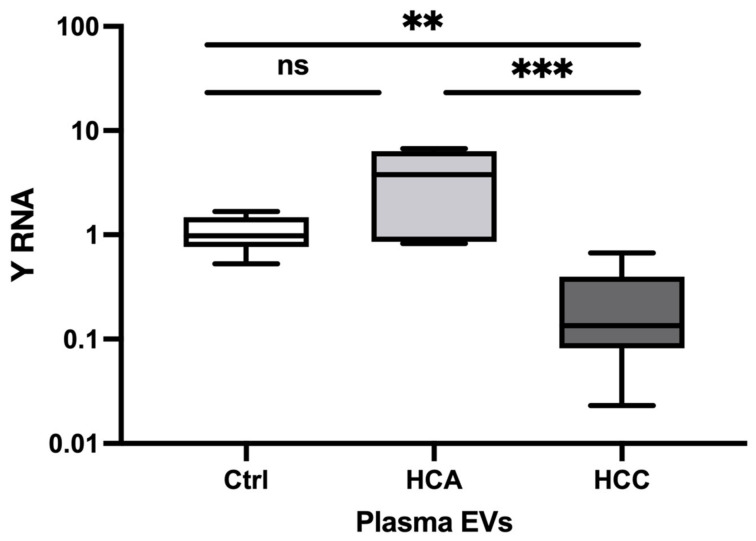
Relative expression of Y RNA in HCA and HCC plasma EV samples. The relative expression level of Y RNA in plasma HCA (*n* = 5) and HCC (*n* = 9) versus Healthy controls (*n* = 6). The *Y*-axis represents relative noncoding RNA expression levels in log10 units. One-Way ANOVA (nonparametric) was performed, followed by the Kruskal–Wallis and Mann–Whitney tests. Differences were considered significant when the *p*-value was < 0.05 (** *p* < 0.01, *** *p* < 0.001). ns: not significant; Ctrl; control, HCA; Hepato-cellular adenoma, HCC; Hepato-cellular carcinoma, EVs; Extra-cellular vesicles.

**Figure 4 animals-13-03054-f004:**
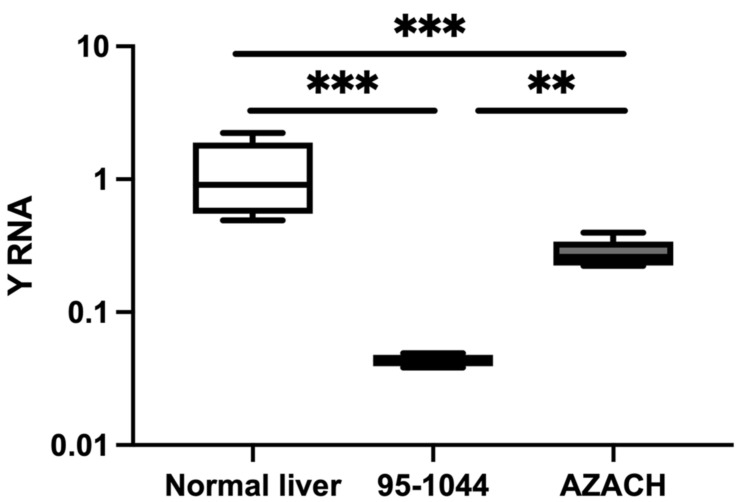
Relative expression of Y RNA in HCC cell lines. The relative expression level in HCC-1044 (*n* = 6) and AZACH (*n* = 6) versus normal liver tissue (*n* = 9). The *Y*-axis represents relative noncoding RNA expression levels in log10 units. The Mann–Whitney U test was performed. Differences were considered significant when the *p*-value was < 0.05 (** *p* < 0.01, *** *p* < 0.001).

**Figure 5 animals-13-03054-f005:**
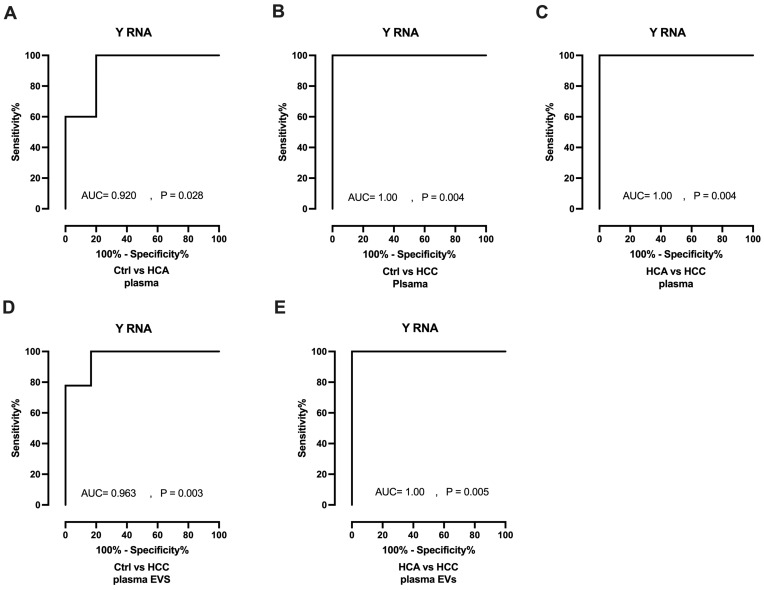
Diagnostic potential of Y RNA as a biomarker. (**A**,**B**). ROC curve (plasma) of Y RNA for differentiating HCA (*n* = 5) and HCC (*n* = 7) group from Healthy controls (*n* = 6). (**C**). ROC curve of Y RNA for differentiating HCC (*n* = 7) from HCA (*n* = 5). (**D**). ROC curve (plasma Evs)) of Y RNA for differentiating HCC (*n* = 9) from Healthy controls (*n* = 6) (**E**). HCC (*n* = 9) versus HCA (*n* = 5). Ctrl: control; HCA: Hepato-cellular adenoma; HCC: Hepato-cellular carcinoma; EVs: Extra-cellular vesicles.

**Table 1 animals-13-03054-t001:** HCA and HCC patient information.

Number	Age	Disease	Sex	Neutered	Breed	Tissue	Plasma
P1	11 Y 3 M	HCA	M	Yes	Crossbreed	15	5
P2	8 Y 1 M	HCA	F	Yes	Miniature dachshund		
P3	10 Y 7 M	HCA	M	Yes	Toy poodle		
P4	12 Y 2 M	HCA	F	Yes	Shiba		
P5	11 Y 6 M	HCA	M	No	Miniature dachshund		
P6	12 Y 3 M	HCA	M	No	Miniature dachshund		
P7	11 Y 9 M	HCA	M	No	Crossbreed		
P8	13 5 M	HCA	F	Yes	Miniature dachshund		
P9	14 Y	HCA	M	No	Golden retriever		
P10	9 Y 2 M	HCA	F	Yes	Toy poodle		
P11	13 Y	HCA	F	Yes	Jack Russell terrier		
P12	11 Y 1 M	HCA	M	No	Miniature dachshund		
P13	12 Y 2 M	HCA	M	No	Crossbreed		
P14	12 Y 3 M	HCA	M	No	Crossbreed		
P15	10 Y 7 M	HCA	M	No	Shiba		
P16	12 Y 3 M	HCC	F	No	Chihuahua	13	9
P17	11 Y 3 M	HCC	F	Yes	Miniature dachshund		
P18	14 Y	HCC	F	Yes	Crossbreed		
P19	10 Y 8 M	HCC	M	Yes	Shiba		
P20	11 Y 7 M	HCC	M	Yes	Welsh corgi		
P21	10 Y 9 M	HCC	F	No	Crossbreed		
P22	10 Y 3 M	HCC	F	No	Beagle		
P23	10 Y 9 M	HCC	F	No	Yorkshire terrier		
P24	11 Y 6 M	HCC	M	No	Shiba		
P25	12 Y	HCC	F	No	Miniature schnauzer		
P26	11 Y 10 M	HCC	M	No	Yorkshire terrier		
P27	13 Y 10 M	HCC	F	No	Shetland sheepdog		
P28	11 Y 7 M	HCC	M	Yes	Crossbreed		

P; Patient, HCA; Hepato-cellular adenoma, HCC; Hepato-cellular carcinoma. F; Female; M; Male.

## Data Availability

All data are provided in the article.

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
