# Peer review of "Novel Y RNA-Derived Fragments Can Differentiate Canine Hepatocellular Carcinoma from Hepatocellular Adenoma"

_animals, 2023, doi:10.3390/ani13193054_

Round 1

Reviewer 1 Report

The study on novel Y RNA-derived fragments as possible diagnostic biomarker in the diagnosis of hepatic tumor is very interesting. Especially because, frequently, hepatic tumors are difficult to diagnose from hepatic biopsies due to their low difference from normal tissue. Thus, any tool helping in this issue is welcome!

The design of the present is good, and the study is well conducted. The introduction is really good (clear and complete). Methods are well described and applied; Results are reliable. Discussion is quite short, but this is the first study on Y RNA in hepatic benign and malignant tumors. In any case the results obtained are clearly commented.

However, before the manuscript may be considered for publication I have a major concern and few minor comments.

MAJOR CONCERN: the number of healthy beagle dog taken as negative control.

As we know, the lines for the use of experimental animals in research, since years, indicate to move basing on the 3R criteria: Reduce, Refine, Replace. In the present study, no problem with refine and replace. My question is: why the authors decided to include 9 normal controls? Why not 5 for example. So I kindly ask authors to motivate the number of the control dog included in the study.

MINOR COMMENTS

Lines 29-30: transfer this sentence at the end of the abstract.

Line 39: change “more in depth” with “further studies are necessary….”

Line 40: hepatocellular neoplastic diseases

Line 52: with male as predisposing factor. This concept is too strong, maybe simple better to say that these tumors are more common in males.

Author Response

First, we appreciate all of kind suggestions. We accept all of your comments and make revised version and specific answer as follows.

Reviewer 1

Response: We thank the reviewer for thoroughly reading our paper and providing such a positive evaluation.

MAJOR CONCERN: the number of healthy beagle dog taken as negative control. As we know, the lines for the use of experimental animals in research, since years, indicate to move basing on the 3R criteria: Reduce, Refine, Replace. In the present study, no problem with refine and replace. My question is: why the authors decided to include 9 normal controls? Why not 5 for example. So I kindly ask authors to motivate the number of the control dog included in the study.

Response: We thank the reviewer for highlighting this issue in our manuscript. We agree with the reviewer's suggestion on the 3R criteria. The control liver samples had been collected from Shin Nippon BioPharma at different times for various projects. Altogether, there were nine samples with reminders available for this study. Our strategy was to increase the control samples (healthy beagle dogs) to strengthen our findings because this is the first discovery about Y RNA in dog liver tissue. As such, we used the maximum number of samples available to carefully investigated the initial experiment. Y RNA showed almost similar expression patterns in the nine individual healthy dogs, which indicated that the Y RNA expression trend was unique in the control group. We concede that 5 or 6 samples would have been fine as the number of negative controls for powering this study, and we will, of course, consider the control sample size based on 3R criteria in our future experiments.

MINOR COMMENTS

Lines 29-30: transfer this sentence at the end of the abstract.

Response: Lines 29-30 have been transferred.

Line 39: change “more in depth” with “further studies are necessary….”

Response: In line 39, the word “more in depth” has been changed to “further”.

Line 40: hepatocellular neoplastic diseases

Response: hepatocellular neoplastic diseases have been added to line 40.

Line 52: with male as predisposing factor. This concept is too strong, maybe simple better to say that these tumors are more common in males.

Response: Line 52 has been revised according to the reviewer’s suggestions.

Reviewer 2 Report

Though poised as a paper describing a method intended to distinguish hepaotocellular adenoma (HCA) from hepatocarcinoma (HCC) in a canine animal model, the Authors astutely touch upon the potential utility in applying the same, potentially lifesaving utility to at-risk humans.  Furthermore, Reviewer finds great interest and merit in exploring cutting edge, molecular tools, including noncoding RNA species overall -- and Y RNA specifically.  

Several issues concern Reviewer, however.  Among these are an apparent failure by Authors to consider androgen-mediated factors as either direct or indirect progenitors of phenotypic shift from HCA to HCC.  While noting a number of canonical pathways triggering inflammatory sequelae, Authors fail to consider the role the androgen steroid hormone markers play in driving cellular conversion from homeostasis to pathogenic proliferation.  

Of perhaps greater concern, and after a broad pubmed.com review of the operative Literature, Reviewer cannot agree that full clarity exists in a grasp of how Y RNA operates -- either in health or disease.  Without citing each paper individually, Reviewer recognizes a strong message that runs like a thread throughout the Literature.  More work -- in fact probably a great deal more work -- shall be required before definitive conclusions may be drawn.  And although Authors do appear to touch upon this, Reviewer perceives that the bold conclusions articulated must be tempered with the vast unknown that presently exists  in comprehending where non-coding RNAs fit in the realm of molecular diagnostic analysis.  

Author Response

First, we appreciate all of kind suggestions. We accept all of your comments and make revised version and specific answer as follows.

Reviewer 2

Though poised as a paper describing a method intended to distinguish hepaotocellular adenoma (HCA) from hepatocarcinoma (HCC) in a canine animal model, the Authors astutely touch upon the potential utility in applying the same, potentially lifesaving utility to at-risk humans.  Furthermore, Reviewer finds great interest and merit in exploring cutting edge, molecular tools, including noncoding RNA species overall -- and Y RNA specifically. 

Several issues concern Reviewer, however.  Among these are an apparent failure by Authors to consider androgen-mediated factors as either direct or indirect progenitors of phenotypic shift from HCA to HCC.  While noting a number of canonical pathways triggering inflammatory sequelae, Authors fail to consider the role the androgen steroid hormone markers play in driving cellular conversion from homeostasis to pathogenic proliferation. 

Of perhaps greater concern, and after a broad pubmed.com review of the operative Literature, Reviewer cannot agree that full clarity exists in a grasp of how Y RNA operates -- either in health or disease.  Without citing each paper individually, Reviewer recognizes a strong message that runs like a thread throughout the Literature.  More work -- in fact probably a great deal more work -- shall be required before definitive conclusions may be drawn.  And although Authors do appear to touch upon this, Reviewer perceives that the bold conclusions articulated must be tempered with the vast unknown that presently exists in comprehending where non-coding RNAs fit in the realm of molecular diagnostic analysis. 

Response: Thanks to the reviewer for highlighting the above points where our manuscript could be strengthened.

Regarding the androgen-mediated factors and androgen steroid hormone, we didn’t investigate or include them in our study design. Our aim was to elucidate whether Y RNA was expressed in hepatocellular carcinoma (HCC) and hepatocellular adenoma tumor (HCA) and whether Y RNA could discriminate HCC from HCA and the control group. This was our first study to identify Y RNA expression in HCC and HCA as a diagnostic biomarker. Therefore, this study didn’t focus on phenotypic shift conversion from HCC and HCA, inflammatory sequelae, and cellular conversion from homeostasis to pathogenic proliferation. We believe that androgen-mediated factors and androgen steroid hormone could be addressed in a future study.

With regard to the 2nd comment, the underlying molecular functions or operation of Y RNA and how it acts on healthy or diseased dogs have yet to be investigated by our group. We discussed this issue briefly in our manuscript as a limitation of our study (lines no. 569 to 570).

We appreciate the reviewer's suggestion on the citation. The citation part has been revised thoroughly in our manuscript.

In our previous study (Reference no. 36), we reported that Y RNA expression was decreased in malignant canine mammary gland tumor (MGT) compared to benign tumors. Then, we hypothesized that Y RNA could be expressed in canine HCC and HCA tumors and could distinguish HCC from HCA. We investigated the expression of Y RNA in clinical HCC and HCA tumor samples in which the expression trend was similar to canine MGT results. We further investigated the Y RNA in HCC cell lines compared to control, which revealed that Y RNA also decreased in HCC cells. Our ROC curves demonstrated a distinct pattern that differed between the HCC and HCA patients, and we believe this is sufficient to draw the current conclusion based on these findings, and we have made the language in the conclusion softer stating “We thus consider that Y RNA has promising potential for differentiating HCC from HCA”.

Our study is the first to investigate Y RNA in hepatic tumors in dogs, including HCC and HCA. While there has been limited research on Y RNA in human cancers, we did briefly discuss the aberrant expression of Y RNA in several types of human cancer in our manuscript. Most research articles suggest Y RNA could be a potential biomarker for cancer. Our findings showed a similarity between decreased Y RNA expression and various diseases, such as canine MGT, human prostate, bladder, and melanoma. After thorough analysis, we strongly believe that Y RNA is a suitable candidate for further investigation as a marker in molecular diagnostic analyses.

Reviewer 3 Report

In the present study, Ushio and colleagues investigated the relative Y-RNA expression levels in dogs with HCC and HCA using qRT-PCR analysis of tumour tissue, plasma and plasma EVs from clinical samples and HCC cell lines to evaluate Y-RNAs as diagnostic biomarkers for these two types of canine liver tumours.

This study may be of interest to readers of this journal, but there are still a few things to consider:

-Figure 4. Please replace 94-1044 with 95-1044 in the graph

- In the statistical analysis section, the authors should indicate whether the graphs or box plots represent the mean ± SD or the mean ± SEM.

- the authors showed that y-RNA is significantly downregulated in 95-1044 cells compared to AZACH cells. They stated that these results are consistent with the results for clinical tumour tissue samples. This is an interesting finding, but could the authors please explain this result in the discussion section?

- I think that the title is not appropriate due to the relatively small number of tissues (and cell lines) analysed. Accordingly, the authors must state in the manuscript that this is a limitation of the study. The present study could be a pilot study.

- Why did the authors decide to use a non-parametric analysis for the statistics?

- Since Y-RNA is down-regulated in rapidly proliferating tissues and cell lines, did the authors include a reference to Y-RNA in metastases?

Author Response

First, we appreciate all of kind suggestions. We accept all of your comments and make revised version and specific answer as follows.

Reviewer 3

In the present study, Ushio and colleagues investigated the relative Y-RNA expression levels in dogs with HCC and HCA using qRT-PCR analysis of tumour tissue, plasma and plasma EVs from clinical samples and HCC cell lines to evaluate Y-RNAs as diagnostic biomarkers for these two types of canine liver tumours.

This study may be of interest to readers of this journal, but there are still a few things to consider:

-Figure 4. Please replace 94-1044 with 95-1044 in the graph

Response: Thanks to the reviewer for highlighting this issue. The cell line name has been replaced in Figure 4 in our manuscript.

- In the statistical analysis section, the authors should indicate whether the graphs or box plots represent the mean ± SD or the mean ± SEM.

 Response: We thank the reviewer’s comments for noticing the statistical analysis section. We represented our data set in Box plots where mean ± SD or mean ± SEM does not fit with applied statistical analysis. Although we decided to use the box plot if the reviewer strongly feels our response is inadequate, we could consider changing the graphics to a scatter plot, where mean ± SD or mean ± SEM could be used; however, we would prefer to keep the existing box plot if at all possible.

- the authors showed that y-RNA is significantly downregulated in 95-1044 cells compared to AZACH cells. They stated that these results are consistent with the results for clinical tumor tissue samples. This is an interesting finding, but could the authors please explain this result in the discussion section?

Response: We appreciate the reviewer's encouraging comments and suggestions for improving our manuscript. We have addressed the highlighted point in the discussion section, with the text below added at lines 519 to 523, “The expression pattern of Y RNA in HCC cell lines was similar to that in clinical tissues. Y RNA was found to be decreased in HCC and in 95-1044 and AZACH cells, versus the control liver samples. This finding is interesting because results for the HCC cell lines reflected those in clinical tissue HCC samples, in comparisons against the same control liver samples”.

- I think that the title is not appropriate due to the relatively small number of tissues (and cell lines) analysed. Accordingly, the authors must state in the manuscript that this is a limitation of the study. The present study could be a pilot study.

 Response: We agree with the reviewer’s comments on sample size. The samples included in this study were relatively small. In light of the reviewer’s comment, we have changed the title and also have softened the conclusion.  We have also added a sentence to the study limitations.

- Why did the authors decide to use a non-parametric analysis for the statistics?

Response: It would have been difficult to analyze our data with a parametric test. Using a nonparametric test with these data was straightforward. Since the nonparametric test only considers the relative ranks of the values, knowing all the values exactly is not essential. In our data, first, we performed 3 normal distribution tests (Anderson-Darling, Shapiro-Wilk, and Kolmogorov-Smirnov) in clinical tissue, plasma, plasma extracellular vesicles, and HCC cell lines. In tissue samples, none of these didn’t pass the normality test. Anderson-Darling didn't pass the normality test for the plasma, plasma extracellular vesicles, and HCC cell lines. For this reason, we decided to perform the non-parametric test.

- Since Y-RNA is down-regulated in rapidly proliferating tissues and cell lines, did the authors include a reference to Y-RNA in metastases?

Response: Yes, we included the reference to the relative expression of Y RNA in metastases. Y RNA was decreased in carcinoma with metastatic mammary gland tumor compared to the benign mixed tumor (Reference no. 36). This finding has been submitted to the peer-reviewed journal and is being under review.

Round 2

Reviewer 3 Report

This manuscript shows major improvements. The authors have addressed my comments/suggestions in a reasonable manner. Thank you.